# Generation of Matrix Degradation Products Using an In Vitro MMP Cleavage Assay

**DOI:** 10.3390/ijms23116245

**Published:** 2022-06-02

**Authors:** Niklas Wagner, Anna E. Rapp, Sebastian Braun, Markus Ehnert, Thomas Imhof, Manuel Koch, Zsuzsa Jenei-Lanzl, Frank Zaucke, Andrea Meurer

**Affiliations:** 1Dr. Rolf M. Schwiete Research Unit for Osteoarthritis, Department of Orthopedics (Friedrichsheim), University Hospital Frankfurt, Goethe University, 60528 Frankfurt am Main, Germany; n.wagner.research@gmail.com (N.W.); rapp@med.uni-frankfurt.de (A.E.R.); zsuzsa.jenei-lanzl@kgu.de (Z.J.-L.); andrea.meurer@kgu.de (A.M.); 2Department of Orthopedics (Friedrichsheim), University Hospital Frankfurt, Goethe University, 60528 Frankfurt am Main, Germany; sebastian.braun@kgu.de (S.B.); markus.ehnert@kgu.de (M.E.); 3Institute for Dental Research and Oral Musculoskeletal Biology, Center for Biochemistry, Faculty of Medicine, University Hospital Cologne, University of Cologne, 50931 Cologne, Germany; thomas.imhof@hditx.de (T.I.); manuel.koch@uni-koeln.de (M.K.)

**Keywords:** MMPs, cartilage, COMP, thrombospondins, osteoarthritis

## Abstract

Matrix metalloproteinases (MMPs) play crucial roles in tissue homeostasis and pathologies by remodeling the extracellular matrix. Previous studies have demonstrated the biological activities of MMP-derived cleavage products. Furthermore, specific fragments can serve as biomarkers. Therefore, an in vitro cleavage assay to identify substrates and characterize cleavage patterns could provide important insight in disease-relevant mechanisms and the identification of novel biomarkers. In the pathogenesis of osteoarthritis (OA), MMP-2, -8, -9 and -13 are of vital importance. However, it is unclear which protease can cleave which matrix component. To address this question, we established an in vitro cleavage assay using recombinantly expressed MMPs and the two cartilage matrix components, COMP and thrombospondin-4. We found a time- and concentration-dependent degradation and an MMP-specific cleavage pattern for both proteins. Cleavage products can now be enriched and purified to investigate their biological activity. To verify the in vivo relevance, we compared the in vitro cleavage patterns with serum and synovial fluid from OA patients and could indeed detect fragments of similar size in the human samples. The cleavage assay can be adapted to other MMPs and substrates, making it a valuable tool for many research fields.

## 1. Introduction

Matrix metalloproteinases (MMPs) are specialized, zinc-dependent proteases which have important functions in physiological processes, such as growth, development, and tissue remodeling and maintenance, but also in pathological conditions [1,2]. Among these conditions are cardiovascular [3,4,5], neurodegenerative [6,7,8] musculoskeletal [9], and renal pathologies [10], as well as cancer [11,12]. Besides their matrix-degrading function, MMPs also release matrix-sequestered growth factors such as latent TGF-beta [13], (in-) activate cytokines [14,15] and process matrix components, thereby generating matrikines, neo-epitopes, and matricryptines, that are then accessible for receptor binding [16,17,18] and might elicit cellular responses in the respective tissue [19,20,21].

Osteoarthritis (OA) is a chronic, low-grade inflammatory disease of the whole joint associated with progressive cartilage matrix degradation leading to an irreversible loss of articular cartilage [22]. MMPs play an important role in OA development [23,24], and their levels increase with the progression of cartilage degeneration [25]. Compared to samples from healthy donors, MMP-2, -8, -9, and -13 were detected in elevated concentrations in synovial fluid and sera from patients with advanced OA [26,27]. Interestingly, their expression by chondrocytes correlates with levels of the cytokines TNFα and IL-1β [28] that can be detected in increased concentrations in osteoarthritic joints.

The major structural components of the extracellular cartilage matrix are collagen II and the proteoglycan aggrecan. Collagen II forms, together with collagen XI and IX, cartilage collagen fibrils and aggrecan assemblies, have gel-like properties. These two suprastructures are interconnected and further stabilized by a large number of small proteoglycans and many non-collagenous glycoproteins, e.g., several members of the thrombospondin family. During OA progression, proteoglycans are released first followed by a degradation of non-collagenous components. However, it is not completely known, which MMPs are able to cleave which cartilage matrix component [29]. Thrombospondin-4 (TSP-4) and cartilage oligomeric matrix protein (COMP or TSP-5) might be MMP substrates in cartilage, as their expression correlates with disease progression. TSP-4 is hardly detectable in healthy adult cartilage, however, its abundance in the cartilage matrix increases with higher OA grades and fragments of TSP-4 have been detected in sera from OA patients [30,31]. In contrast to TSP-4, COMP is an integral part of the cartilage matrix and COMP can be detected in large amounts already in healthy cartilage. However, it is degraded in early OA but re-expressed in later stages [31,32]. COMP is widely used as an OA biomarker and its serum concentration correlates with both OA severity and the number of affected joints [33,34]. Most ELISAs used for COMP detection do not distinguish between full-length and degraded COMP, but specific COMP fragments have also been detected in the sera of OA patients [34].

It has been shown previously that proteases of the a disintegrin and metalloprotease with thrombospondin motifs (ADAMTS)-family can cleave COMP in vitro, yielding fragments that could also be detected in OA synovial fluid [35]. Even though the in vitro cleavage of COMP by different MMPs has been demonstrated, the relevance of this cleavage for OA has not been studied systematically. Other matrix components, including aggrecan and fibronectin, are processed by MMPs and/or ADAMTs. This cleavage led to the generation of biologically active fragments such as the 32-mer of aggrecan implicated in OA pain [36] and fragments of fibronectin [20,37], some of which are signaling via toll-like receptor 2 [38,39]. In summary, there is still a lack of knowledge on the topic of MMP-generated cartilage ECM fragments. However, a better understanding could help to identify both novel targets for the treatment and biomarkers of OA.

Therefore, we aimed to set up an in vitro MMP cleavage assay and studied the degradation of two cartilage matrix components, COMP and TSP-4, by the OA relevant MMPs 2, -8, -9, and -13. We could show that both recombinant proteins are cleaved in a time- and concentration-dependent manner. The resulting fragments can be enriched, purified, and used in cell-based assays to analyze their bioactivity. Finally, we identified similar fragments in synovial fluid and serum of OA patients further strengthening the importance of MMP-mediated cartilage degeneration in OA.

## 2. Results

### 2.1. Substrates and MMPs Used for Digestion Assays

First, we expressed COMP, TSP-4 and different MMPs recombinantly and purified all proteins by affinity chromatography in order to use them as substrates and enzymes in digestion assays. To verify the expression of full-length proteins and demonstrate the homogeneity of the purified proteins, the final preparations were resolved on a 10% polyacrylamide gel under reducing conditions and stained with Coomassie Brilliant Blue G250. For COMP, a single band with an apparent molecular weight of approximately 110 kDa could be detected, as expected. TSP-4 showed a double band with molecular weights of approximately 155 and 135 kDa (Figure 1A). The appearance of this doublet could either be due to different post-translational modifications or to a proteolytic cleavage, as reported earlier for thrombospondins [40]. All MMPs could also be expressed and purified as inactive pro-forms in sufficient amounts and integrity. Their activatability could be demonstrated after incubation with 1 mM APMA for 2 h, as indicated by a shift towards a lower molecular weight representing the active forms (Figure 1B).

### 2.2. MMPs Cleave COMP and TSP-4 in a Concentration-Dependent Manner

The incubation of COMP and TSP4 with different concentrations of MMP-2, -8, -9, and -13 for 24 h revealed a concentration-dependent cleavage of both substrates. Incubation of COMP with higher concentrations of MMP-2 (5 and 0.5 µg/mL) resulted in a shift in molecular weight from the full-length band of 110 kDa to approximately 90 kDa and the appearance of additional bands at 70 and 60 kDa as visualized by RotiBlue^®^ staining (Figure 2A upper panel) and Western Blot (Figure 2A lower panel). Lower concentrations (5 × 10^−2^ to 5 × 10^−6^ µg/mL) did not cleave COMP efficiently. The same observation was true for MMP-8 and -9. However, for these MMPs, the cleavage was most efficient in the highest MMP to substrate ratio compared to the second dilution step (Figure 2B,C). Only for MMP-13, efficient cleavage of the substrate was detectable for the three highest concentrations (5 µg/mL, 0.5 µg/mL, and 5 × 10^−2^ µg/mL), evident by the reduced size of the full-length band (Figure 2D). In the case of digestion with MMP-13, the band of approx. 60 kDa was more prominently visible in comparison to the other MMPs. Efficient cleavage was only observed when MMPs where activated with AMPA but not without addition of the activator (Appendix A).

Cleavage of TSP-4 by the different MMPs resulted in a slightly different pattern. Efficient cleavage for MMP-2, and -9 was observed when 5 µg/mL MMP were used, indicated by a reduction of the molecular weight and bands at approximately 110 and 80 kDa (Figure 3A,C). Absence of the full-length double band and a reduction in size was also observed for 0.5 µg/mL MMP, and resulted in bands at approximately 120 and 90 kDa, though less pronounced.

For MMP-8, cleavage was observed when 5 µg/mL MMP was used, leading to bands at approximately 110 kDa and for 0.5 µg/mL MMP with a slightly different pattern, and a band at 120 kDa (Figure 3B).

Similar to the concentration-dependent digestion of COMP, MMP-13 was able to cleave TSP-4 in a wider concentration range between 5 µg/mL and 5 × 10^−3^ µg/mL (Figure 4D). Fragments were observed at 110, 90, and 80 kDa, while the 80 kDa fragment was present with higher MMP concentrations only (Figure 3D).

In summary, cleavage of COMP and TSP4 was observed for all MMPs, with slight differences regarding the active concentrations. MMP-13 was able to cleave both substrates over the widest concentration range, and a cleavage could be observed already with 5 ng/mL. For the other MMPs, only the two highest concentrations used resulted in a detectable proteolysis.

### 2.3. Kinetics of COMP- and TSP-4-Cleavage by MMPs

To determine the dynamic cleavage of the substrates by MMPs, we performed a kinetic analysis and analyzed the cleavage after 2 h, 6 h, and 24 h (Figure 4). We used a concentration (0.5 µg/mL) in which good cleavage was observed in the concentration-dependent assay (Figure 2 and Figure 3). While MMP-2, -8, and -9 show only minor cleavage after 2 h, MMP-13 already shows a strong reduction of the full-length signal at this timepoint for both substrates (Figure 4B,D). Cleavage of the substrates by MMP-2, -8, and -9 proceeds continuously over the next hours until the endpoint. MMP-13-mediated cleavage almost reaches its endpoint at 6 h incubation, after which cleavage only progresses further slowly. Generally, MMP-8 seems to be the least efficient MMP regarding cleavage of COMP and TSP-4.

### 2.4. Physiological Concentrations of MMP Lead to Stable Cleavage Products in Long-Term Digestion Assay

To investigate if MMPs generate stable fragments, we added MMP-2, 8, 9, and -13 in two concentrations (indicated in Figure 9 in Materials and Methods) to the substrates COMP and TSP-4 and incubated the solution for two weeks. During this time, no additional MMP was added. Incubation of COMP with the higher MMP concentrations resulted in a reduction in size of the full-length band to approximately 90 kDa and a new band at around 70 kDa for all MMPs (Figure 5A). Lower MMP concentrations resulted in a strongly reduced cleavage of COMP. In immunoblots, faint bands at approximately 70 kDa could be observed for MMP-2 and MMP-9. These bands were hardly visible after incubation with MMP-8 and MMP-13 (Figure 5B).

High concentrations of MMP-2, -9, and -13 resulted in the complete absence of the doublet band for TSP-4 and a shift in molecular weight. Three distinct bands could be observed at approximately 110, 90, and 80 kDa. High concentrations of MMP-8 induced a reduction in the size of the full-length bands, further bands were hardly visible (Figure 6A). Lower MMP concentrations were not able to induce efficient cleavage, although a shift in size of the full-length band could be observed for MMP-2 and MMP-13 (Figure 6B).

### 2.5. Stable COMP and TSP-4 Cleavage Products Could Be Detected in Human Synovial Fluid and Serum Samples

To investigate whether MMP-generated cleavage products are present in vivo, synovial fluid and serum samples from OA patients and healthy controls were analyzed using immunoblots. The resulting band pattern was compared with bands and fragments after long-term digestion of recombinant COMP and TSP-4 with higher MMP concentrations (Figure 7 and Figure 8).

A COMP cleavage product of 90 kDa could be detected in all synovial fluids of OA patients. In addition, a smaller 70 kDa COMP fragment was found. One synovial fluid sample displayed a further prominent band at approx. 30 kDa (Figure 7A). The 90 kDa fragment could also be detected in serum, irrespective of whether the serum was from OA patients or healthy controls (Figure 7B). Interestingly, a prominent fragment of 40 kDa was present in all OA serum samples, while it was absent or strongly reduced in most healthy samples (Figure 7B).

TSP-4 cleavage products with the size of approx. 110 kDa were detected in synovial fluids (Figure 8A) and in serum samples (Figure 8B). Furthermore, in serum samples of OA patients, more intense signals compared to healthy controls are visible. Smaller 80 kDa cleavage products were detectable in synovial fluid and serum samples, but also difference in serum of healthy controls (Figure 8A,B).

In summary, we could show that samples from OA patients but also from healthy controls contain fragments of both COMP and TSP-4. The similar size of distinct fragments might indicate that MMPs are also responsible for COMP and—to a lesser extent—TSP-4 fragmentation in vivo.

## 3. Discussion

MMPs play an important role in physiological tissue remodeling but also in degenerative processes and disease development and progression [1,2,3,4,5,6,7,8,9,10,11,12,23,24]. In OA, MMP levels increase with the progression of the disease [25] and a larger number of cartilage ECM components are cleaved by MMPs [9]. The degradation of the cartilage matrix follows a specified sequence with proteoglycans being degraded first, followed by the cleavage of minor components and finally cartilage collagen fibrils [22]. Cleavage products are released into the synovial fluid and eventually can be detected in the serum where they might serve as biomarkers for OA. In previous studies, a number of different MMPs in a wide range of concentrations were detected in the synovial fluid and serum of OA patients [26,27,41,42,43,44]. In the present study, we selected MMP concentrations based on a literature research. However, it could be interesting to determine the actual MMP concentration in individual samples and to correlate these concentrations with fragmentation patterns. However, this should be done immediately after sample collection to avoid activity loss or self-digestion. It is not completely understood which MMPs are able to cleave which ECM components, and whether the cleavage products are physiologically relevant and might contribute to the progression of OA. For MMPs, a real consensus sequence that would allow a prediction of substrate cleavage does not exist. Therefore, we set up an in vitro digestion assay using recombinantly expressed MMPs and ECM components in which we could analyze the time- and concentration-dependent generation of cleavage products.

Based on the current literature, we used the MMPs-2, -8, -9, and -13 in a concentration range that has been detected in patient samples (see Figure 9). As substrates, we chose the two cartilage matrix glycoproteins, COMP and TSP-4. Their functional roles and relevance in OA have been described earlier [31] but their cleavage by MMPs has not been studied systematically yet.

To identify effective MMP concentrations, we first used a dilution series and incubated the MMPs with their substrates for 24 h. Even though specific MMP concentrations have been reported in patient samples, it is not known how long the proteases interact with their substrates in vivo. Therefore, the results of this short-term assay have to be interpreted with caution. We also investigated the kinetics of cleavage for up to 24 h. A substantial cleavage could be observed already after 6 h for MMP-13, and after 24 h for all MMP/substrate combinations. As the concentrations are close to the (patho-) physiological range, we are confident that the observed cleavage could reflect the in vivo situation. In addition, cleavage fragments of similar size have been reported in previous studies.

Using COMP as a substrate, we could identify similar fragment patterns for all MMPs: prominent bands appeared at approximately 90 and 70 kDa, and a weaker fragment at 60 kDa. All MMPs were able to efficiently cleave the substrate in the highest concentration, resulting in all three bands. In lower concentrations, however, the efficiency varied with MMP-13 seemingly the most effective protease over the concentration range. Two major MMP-13-generated fragments of COMP with approximately 85 and 50 kDa and an MMP-9-generated fragment of about 90 kDa were reported previously, supporting the validity of our approach. In contrast to a previous study, in which cartilage tissue was digested with different proteases [45], we found that MMP-2 and -8 were able to cleave COMP efficiently. We found that the products of 70 and 60 kDa were not described in the study mentioned above, however, others observed similar fragments [46,47]. The fact that these cleavage products could not be detected in cartilage tissue could be explained either by a limited accessibility or by a complete degradation of the substrate in vivo. More studies are needed to determine if these fragments are relevant and/or biologically active.

As mentioned before, the interaction time of enzyme and substrate during progressive cartilage degeneration in vivo is not known. To consider and approximate a slow turnover, we used a long-term incubation with physiological MMP concentrations. This approach should also lead to the identification of fragments that remain stable even after prolonged exposure to MMPs. The incubation for 2 weeks resulted in 90 and 70 kDa fragments for all MMPs in high concentrations, indicating that these fragments are refractory to further cleavage. This is further supported by the presence of similar fragments in synovial fluid of OA patients. In serum and synovial fluid, we found an additional immunoreactive fragment of approximately 40–50 kDa, suggesting that there are further COMP-digesting proteases active in OA, among them other MMPs and/or members of the ADAMTS-family. ADAMTS-7 and -12 have been shown to interact with and to cleave COMP. Interestingly, the levels of both enzymes were also increased in arthritic diseases [48]. As expected, lower MMP concentrations resulted in less efficient cleavage. It is unknown how long the MMPs remain active over the incubation of 2 weeks. However, comparing identical concentrations in short- versus long-term assays leads to the assumption that the activity lasts definitely longer than 24 h because the ratio of the fragment bands is different. In tissue, MMPs are produced continuously upon a certain stimulus, and the activity of an individual MMP molecule might depend on many factors. Therefore, our assay might not completely mimic the (patho-)physiological situation in the tissue but still gives important insight into the potential of different MMPs in generating cartilage matrix protein fragments.

COMP and TSP-4 are closely related and share an overall 46% sequence identity with much higher homology in distinct domains [49] (sequence alignment displayed in Appendix A). Therefore, a similar cleavage pattern was not surprising, and the fragmentation of TSP-4 after digestion with the MMPs generally resembled that of COMP. However, the exact cleavage sites have to be determined in future studies. We detected major TSP-4 fragments of approximately 120 kDa and 80–90 kDa, depending on the used MMP. Similar to COMP, the most efficient cleavage was observed for the highest MMP-concentration, and MMP-13 was cleaving TSP-4 over the widest concentration range. The long-term incubation resulted in 3 distinct TSP-4 fragments for all MMPs, with MMP-8 being the least efficient. When lower MMP concentrations were used, cleavage was less efficient and full cleavage was not observed before the MMPs lost activity. To our knowledge, cleavage of TSP-4 by OA-relevant MMPs has not been investigated previously. However, as TSP-4 expression correlates with OA severity, its cleavage might be relevant. COMP and TSP-4 share structural similarities, but their expression in healthy and osteoarthritic cartilage as well as their biological function differ [31], and at present, the potential biological activity of fragments remains largely speculative. However, a previous study could show that the level of specific TSP-4 fragments is increased in serum samples from OA patients [50]. Further, full-length TSP-4 has been shown to be upregulated in different injury models, leading to neuropathic pain [51] and a role for TSP-4 in promoting spinal sensitization after painful mechanical joint injury has been described [52]. Blocking or genetic inactivation of TSP-4 prevented hypersensitivity, while intrathecal injection caused it and increased the frequency of postsynaptic potentials [53]. TSP-4 cleavage products might also be involved in such processes; however, further analysis is needed. The identification of OA-specific COMP or TSP-4 fragments generated by OA-specific proteases might help to improve the prognostic value of serum-based biomarker assays. The use of cleavage products rather than the full-length protein as a biomarker has been suggested for type 2 [18,34,54,55]. The use of specific software and artificial intelligence for prediction cleavage products and to perform protein-protein-docking could possibly speed up the search for COMP- and TSP-4-derived matrikines or biomarkers. However, such in silico investigations are not possible unless accurate and validated 3D-structures of the pentameric proteins are available, which is unfortunately not yet the case. Currently, the crystal structures have been solved only for smaller fragments and not the full-length proteins. In addition, the assembly of full-length subunits into oligomeric structures has to be considered to predict the accessibility of proteases to their substrates. AlphaFold structures of monomeric COMP from different species are available (https://alphafold.ebi.ac.uk/ accessed on 24 May 2022). However, pentameric COMP alone consists of more than 3500 amino acids, and when taking calcium-dependent conformational changes into account, the calculation time for such predictions would be immense. Stable protein complexes have been resolved. However, at this stage, predictions on huge protease/substrate complexes remain rather speculative and, so far, protease cleavage sites have not yet been identified with AlphaFold, even though the accuracy of modelling larger structures using structure prediction is steadily increasing.

Matrikines of other ECM components were already shown to be biologically active in OA-relevant tissues and cells. Injection of a 32-peptide fragment of aggrecan generated by MMP and ADAMTS activity into the knee of wildtype mice resulted in hyperalgesia mediated by TLR-2, while blockade of the 32-mer production in vivo protected against early knee hyperalgesia in a model for OA [38]. Another example is the 29 kDa fragment of fibronectin, which was shown to promote proteoglycan release form cartilage cultures [56] and modulates the expression of MMPs in chondrocytes via TLR-2/TonEB [39]. Fragments of collagen II and of matrilin-3 have been used to stimulate primary chondrocytes and induce the expression of inflammatory cytokines and proteolytic enzymes that could contribute to a perpetuation of cartilage degeneration [57]. The same group used COMP constructs based on the protein domain structure. However, these fragments were not able to modify the expression of genes involved in cartilage homeostasis [58]. The authors discussed that the activity of fragments generated by MMPs might be different. This can now be investigated in more detail with fragments derived from our cleavage assay. A mass spectrometric analysis identified three COMP-derived peptides that are only present in OA cartilage [58]. It is unclear whether these peptides were generated by MMPs, but so far, no biologic activity on endothelial cells and synovial fibroblasts could be demonstrated and it remains to be determined if these peptides could influence chondrocytes [58].

In summary, we present a versatile and quite easily adaptable in vitro assay for the digestion of (OA-relevant) ECM components with disease-associated MMPs in (patho-) physiological concentrations. We established our assay using recombinant COMP and TSP-4 and were able to show, first, that all MMPs are able to cleave both proteins. Second, we found similar cleavage products in both synovial fluid and serum from OA patients, thereby indicating the reliance of the assay to generate physiological products. The system can be easily adapted to further substrates and effector proteases relevant in OA or other conditions and diseases of the ECM. The assay can serve as a first step to identify novel cleavage products that can then be further characterized and used to investigate their biological activity in different cell types and tissues and maybe even to generate fragment-specific diagnostic tools.

## 4. Materials and Methods

### 4.1. Recombinant Expression and Protein Purification

The constructs used in the present study were generated by cloning the full-length sequences of human COMP (Gene ID 1311), human TSP-4 (THBS4, Gene ID 7060) and murine MMP-2 (Gene ID 17390), -8 (Gene ID 17394), -9 (Gene ID 17395) and -13 (Gene ID 17386) into an eukaryotic expression vector based on the plasmid pCEP-Pu, as described earlier [31,59]. In brief, the full-length sequences were amplified by PCR. All PCR products were lacking the natural signal peptide sequences and harbored a NheI restriction site at the 5′-end and a XhoI site at the 3′-end. After digestion with NheI and XhoI, the amplified cDNAs were cloned into the modified episomal expression vector pCEP-Pu in-frame with the 5′ sequence of the BM-40 signal peptide. Thus, all constructs carried the BM-40 signal peptide to promote secretion of the proteins and a double strep-II tag [60] to facilitate protein purification. HEK293 EBNA [61] cells were transfected with these constructs using FuGene HD according to the manufacturer’s instructions (Promega, Madison, WI, USA). Transfected cells were selected by puromycin application for 14 days (0.5 µg/mL, Sigma-Aldrich, St. Louis, MO, USA) and cultured in DMEM/F-12 (Gibco, ThermoFisher Scientific, Waltham, MA, USA) with 1% FBS (Sigma-Aldrich, St. Louis, MO, USA). After selection, supernatants were collected every second day over a period of 2 weeks, immediately supplemented with PMSF (Carl Roth, Karlsruhe, Germany) and NEM (Sigma-Aldrich, St. Louis, MO, USA) to a final concentration of 1 mM and cleared by centrifugation. For protein purification, supernatants were run over Strep-Tactin XT columns following the manufacturer’s instructions (IBA Lifescience, Goettingen, Germany). Protein concentrations in the eluate were determined using the Qubit ProteinAssay Kit (ThermoFisher Scientific, Waltham, MA, USA). All proteins were stored at −80 °C until use.

### 4.2. In Vitro Activation of MMPs

Purified MMPs were activated with 1 mM 4-aminophenylmercuric acetate (APMA) (Sigma-Aldrich, St. Louis, MO, USA) in digestion buffer (TCNB, 50 mM Tris pH 7.5 (neoFroxx, Einhausen, Germany), 10 mM CaCl_2_, 150 mM NaCl, 0.05% Brij-35 (all Sigma-Aldrich, St. Louis, MO, USA)) at 37 °C [62]. When digesting matrix proteins, MMPs and matrix proteins were first mixed in the ratio indicated in TCNB, then 1 mM APMA was added for activation.

### 4.3. Physiological MMP Concentration Ranges in Synovial Fluid

A literature review was performed to identify physiological ranges of OA-relevant MMPs [26,27,41,42,43,44]. Concentration ranges and concentrations used for each MMP in the digestion assays are depicted in Figure 2. For the long-term cleavage assay, higher (marked with ‘I’ in the bar) and lower (marked with ‘II’) concentrations were used. “Lower” represents the average of the concentration reported in different studies, “higher” represents a ten times higher but still physiological concentration.

### 4.4. Digestion of COMP and TSP-4 with MMPs

In short-term digestion assays, 1 µg of purified TSP-4 or COMP, respectively, was digested with MMPs in concentrations ranging from 5 µg/mL to 5 × 10^−5^ µg/mL in 10-fold dilution steps in TCNB in a total volume of 10 µL. MMPs were activated as described above and the mixtures were incubated at 37 °C for 24 h. Substrate without MMPs, and activated MMPs without substrate (5 µg/mL) were used as controls. To analyze the cleavage kinetics, 1 µg of purified COMP and TSP-4, respectively, were mixed with MMPs in TCNB at a concentration of 0.5 µg/mL. MMPs were activated with 1 mM APMA as described above. Reactions were stopped after 2, 6, and 24 h. Substrate without MMP addition was used as a control. In long-term digestion assays, 1 µg of each substrate was digested with two MMP concentrations in a (patho-) physiological range described earlier and as summarized in Table 1. The MMPs were activated at the beginning of the incubation period as described above and incubated at 37 °C for 14 days without further addition of MMPs. Ranges of these studies and used MMP concentrations are shown in Figure 9.

After the respective incubation period, all samples were stored at −20 °C until further analysis.

### 4.5. Harvesting of Synovial Fluid and Serum Samples

Blood and synovial fluid were collected from patients undergoing total knee replacement surgery at the Department of Orthopedics (Friedrichsheim), University Hospital Frankfurt, after obtaining informed consent (Ethics committee University Hospital Frankfurt, vote no. 19-347). After clotting at 4 °C, serum was collected by centrifugation at 3500× *g*. Synovial fluid was aliquoted without further processing. Serum samples from voluntary donors without OA diagnosis served as “healthy” controls. Serum and synovial fluid samples were stored at −80 °C until further analysis.

### 4.6. SDS-Polyacrylamide Gel Electrophoresis and Immunoblot Analysis

Recombinant proteins, protein digests, synovial fluid and serum samples were mixed with 4x sample buffer (NuPAGE LDS Sample Buffer (4x), ThermoFisher Scientific, Waltham, MA, USA) containing 10% 2-mercaptoethanol (Carl Roth, Karlsruhe, Germany) and resolved on 10% SDS-polyacrylamide gels (Carl Roth, Karlsruhe, Germany) at 150 V for 75 min. To visualize whole protein, 1% trichloro-ethanol (Sigma-Aldrich, St. Louis, MO, USA) was added to the resolving gel, which was activated by UV for 1 min after separation [63,64] on a ChemiDoc™ XRS+ (Bio-Rad, Munich, Germany). Proteins were blotted onto a 0.45 µm polyvinylidene fluoride (PVDF) membrane (Amersham™ HyBond™, GE Healthcare, Munich, Germany) at 200 mA for 1 h. Membranes were blocked with 5% skim milk (blotting grade, Carl Roth, Karlsruhe, Germany) in TBS pH 7.6 with 0.1% Tween-20 (TBST) for 1 h at room temperature and with gentle agitation. The membranes were incubated for 16 h at 4 °C with polyclonal sera directed against TSP-4 [65] raised in a rabbit and guinea pig, and the COMP 4-1 fragment [66] raised in a rabbit, respectively. To detect cleavage products in kinetic experiments, membranes were incubated with StrepTactin-HRP (IBA Lifescience, Göttingen, Germany) overnight.

After washing with TBST, the membranes were incubated with the appropriate HRP-coupled secondary antibody for 1 h at RT (Jackson ImmunoResearch, Ely, UK or DAKO Agilent, Santa Clara, CA, USA). Antibody binding was visualized by enhanced chemiluminescence using 100 mM Tris-HCl pH 8.5, 225 mM p-coumaric acid, 1.25 mM luminol (Sigma-Aldrich, St. Louis, MO, USA), and 2.94 mM H_2_O_2_ (Sigma-Aldrich, St. Louis, MO, USA). The signal was detected using Chemi Doc™ XRS+ (Bio-Rad, Munich, Germany) molecular imager and the ImageLab™ software (version 5.1, Bio-Rad, Munich, Germany) in signal accumulation mode.

To detect all MMP-generated fragments, gels were stained with 1x Roti^®^Blue Protein Staining Solution (colloidal Coomassie^®^ brilliant blue G250; Carl Roth, Karlsruhe, Germany) over night at RT and with agitation. The intensity and molecular weight of the fragments were determined using ImageLab™ with Precision Plus Protein Dual Color Standard (BioRad, Munich, Germany) as a reference. Graphs were created using GraphPad Prism 9 for macOS (version 9, GraphPad Software, San Diego, CA, USA).

## Figures and Tables

**Figure 1 ijms-23-06245-f001:**
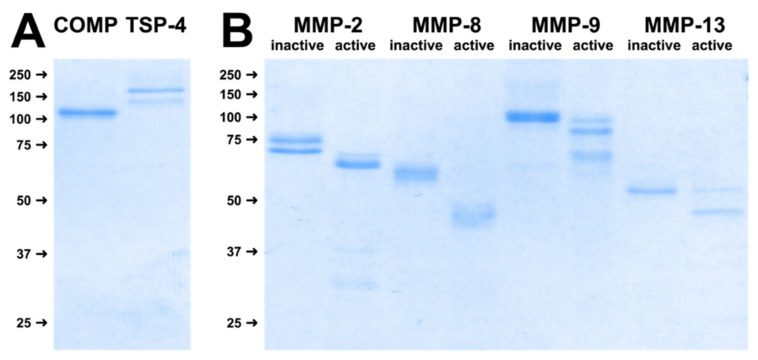
1 µg of purified COMP and TSP-4 were resolved on a 10% gel under reducing conditions and visualized by Coomassie staining (**A**). Recombinantly expressed and purified matrix metalloproteinases (MMPs) were activated by the addition of 1 mM 4-aminophenylmercuric acetate (APMA). Incubation of the inactive pro-forms with APMA resulted in a shift towards smaller proteins representing the activated forms of the corresponding MMP (**B**).

**Figure 2 ijms-23-06245-f002:**
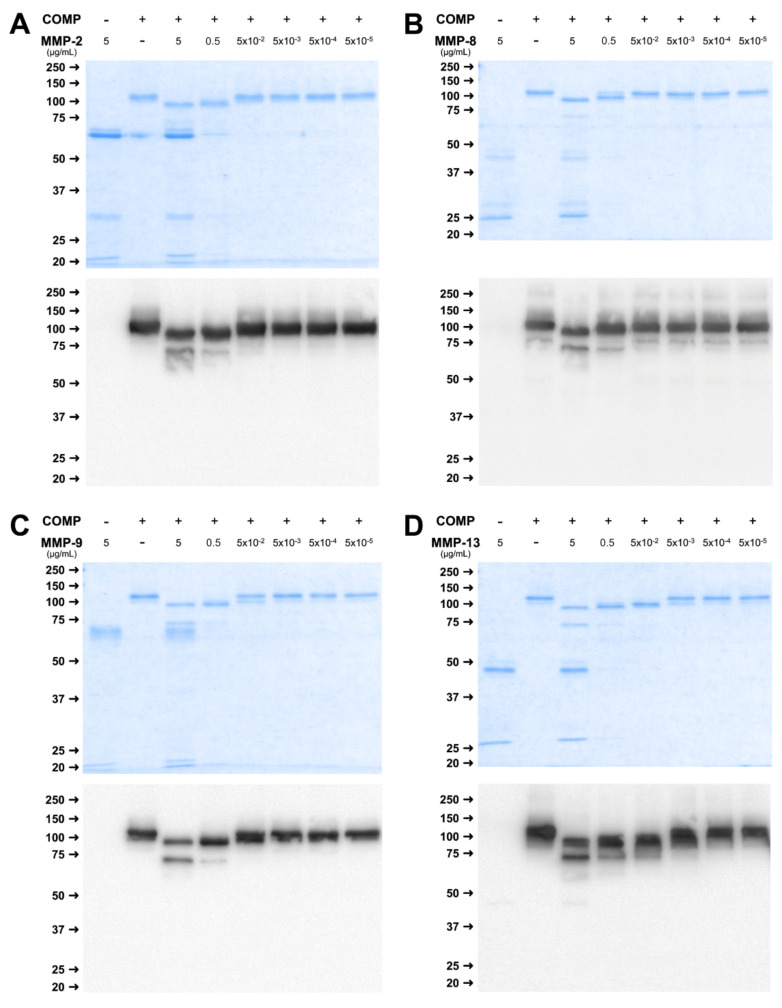
MMPs cleave COMP in a concentration-dependent manner. COMP was incubated with decreasing concentrations of MMP-2 (**A**), -8 (**B**), -9 (**C**), and -13 (**D**) for 24 h and cleavage was visualized by RotiBlue^®^ staining and Western Blot using a polyclonal COMP antibody. Protein sizes are shown in kDa, MMP concentrations in µg/mL.

**Figure 3 ijms-23-06245-f003:**
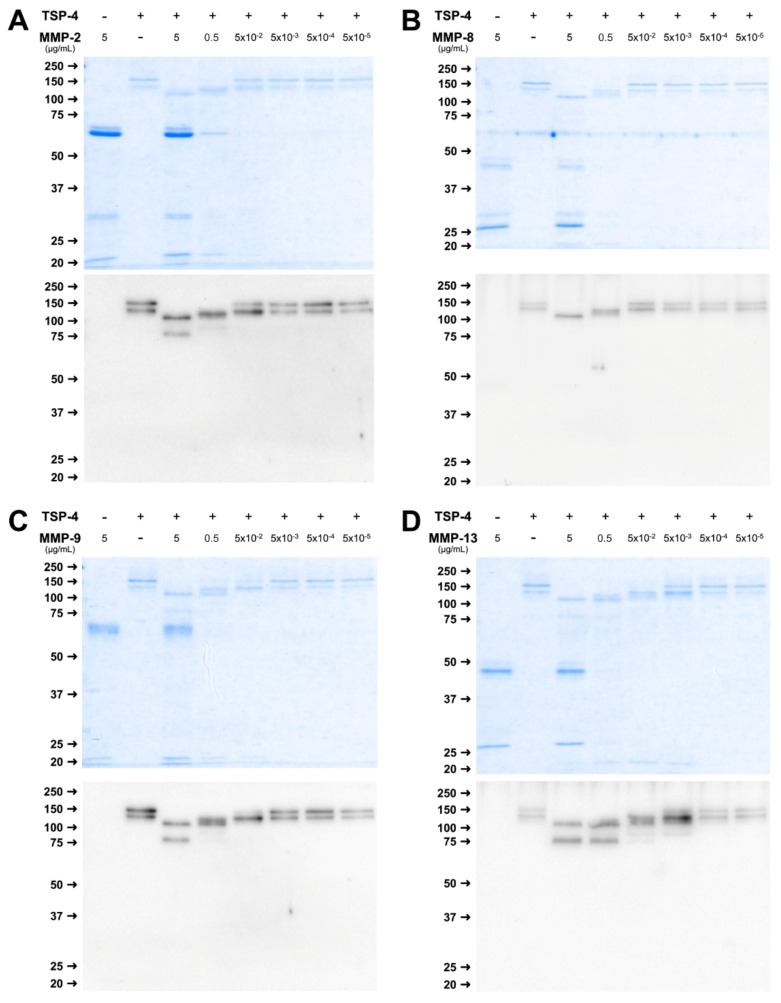
MMPs cleave TSP-4 in a concentration-dependent manner. TSP-4 was incubated with decreasing concentrations of MMP-2 (**A**), -8 (**B**), -9 (**C**), and -13 (**D**) for 24 h. Fragments were visualized by RotiBlue^®^ staining and Western Blot using a polyclonal TSP-4 antibody. Protein sizes are shown in kDa, MMP concentrations in µg/mL.

**Figure 4 ijms-23-06245-f004:**
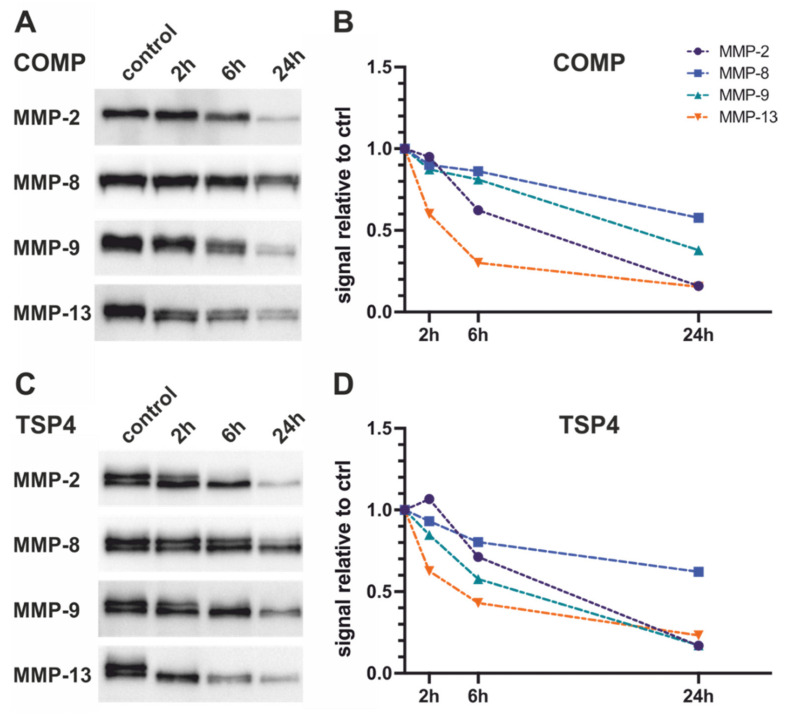
Kinetics of COMP- (**A**,**B**) and TSP-4- (**C**,**D**)-cleavage by MMPs over 24 h. Substrates were detected by staining of the strep-II tag using StrepTactin^®^-HRP (**A**,**C**). Relative changes in the detected signal normalized to the untreated control are depicted in (**B**,**D**).

**Figure 5 ijms-23-06245-f005:**
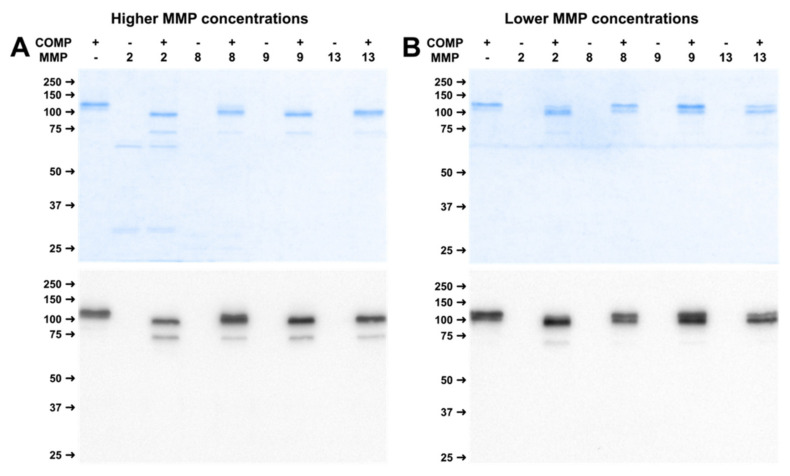
Long-term digestion of COMP with MMP-2, -8, -9 and -13 for two weeks results in stable cleavage products. (**A**) shows physiologically higher, (**B**) physiologically lower concentrations. Roti^®^Blue protein staining and Western Blot using a polyclonal COMP antibody were utilized for cleavage analysis. ‘+’ indicates with, ‘−‘ without substrate or MMP. Protein sizes are shown in kDa.

**Figure 6 ijms-23-06245-f006:**
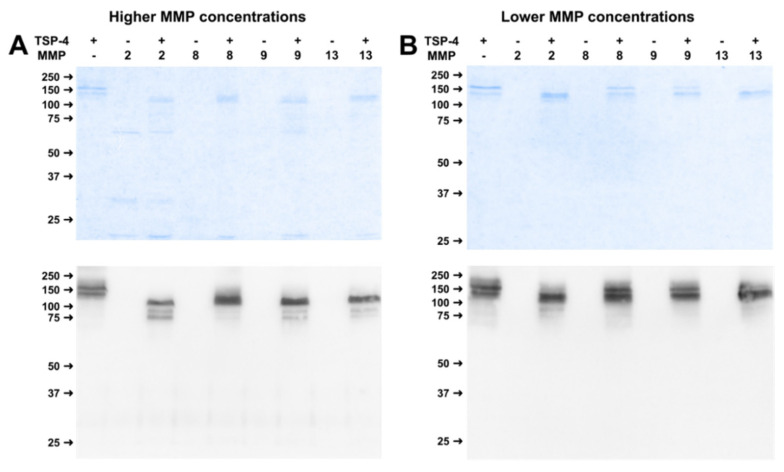
Long-term digestion over two weeks of TSP-4 with MMP-2, -8, -9, and -13 resulted in stable cleavage products. (**A**) shows physiologically higher, (**B**) physiologically lower concentrations. Roti^®^Blue protein staining and western blot using a polyclonal TSP-4 antibody was utilized for cleavage analysis. ‘+’ indicates with, ‘−‘ without substrate or MMP. Protein sizes are shown in kDa.

**Figure 7 ijms-23-06245-f007:**
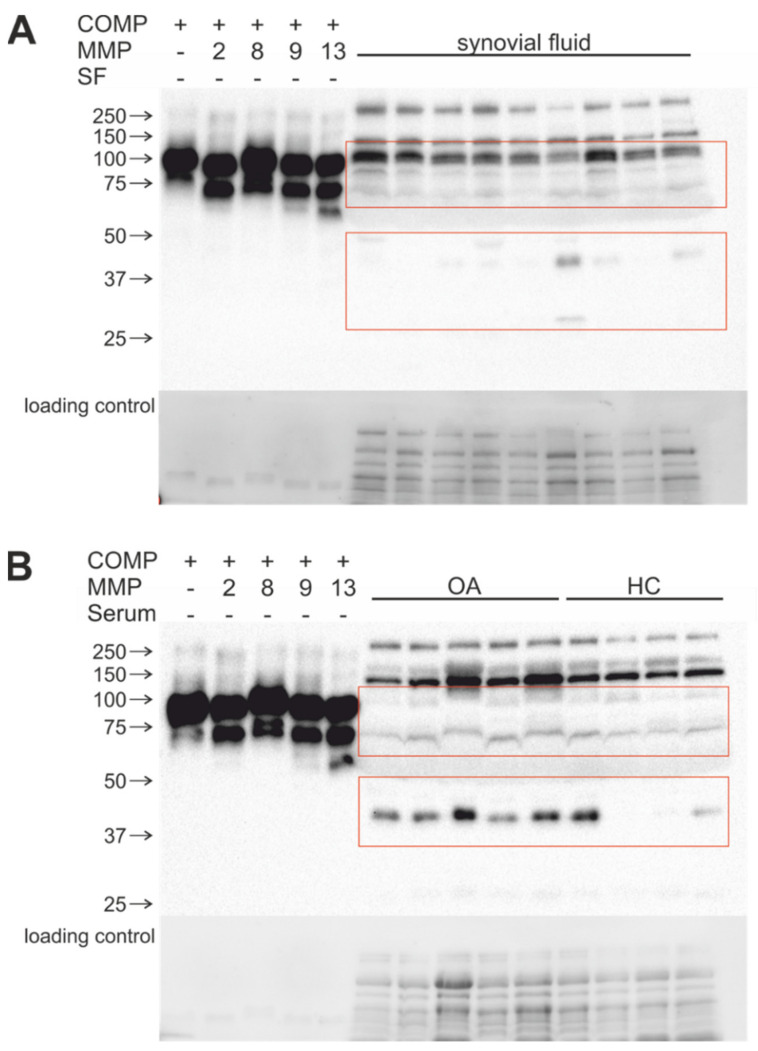
Comparison of in vitro MMP-generated COMP fragments with cleavage products detected in human synovial fluid (**A**) or serum samples (**B**). In (**B**) serum samples from healthy individuals (HC) were included. Fragments were visualized with a polyclonal COMP antibody. Red boxes indicate relevant protein fragments. (+) indicates with, (−) without substrate; protein sizes are shown in kDa. The lower panels in (**A**,**B**) display the loading control.

**Figure 8 ijms-23-06245-f008:**
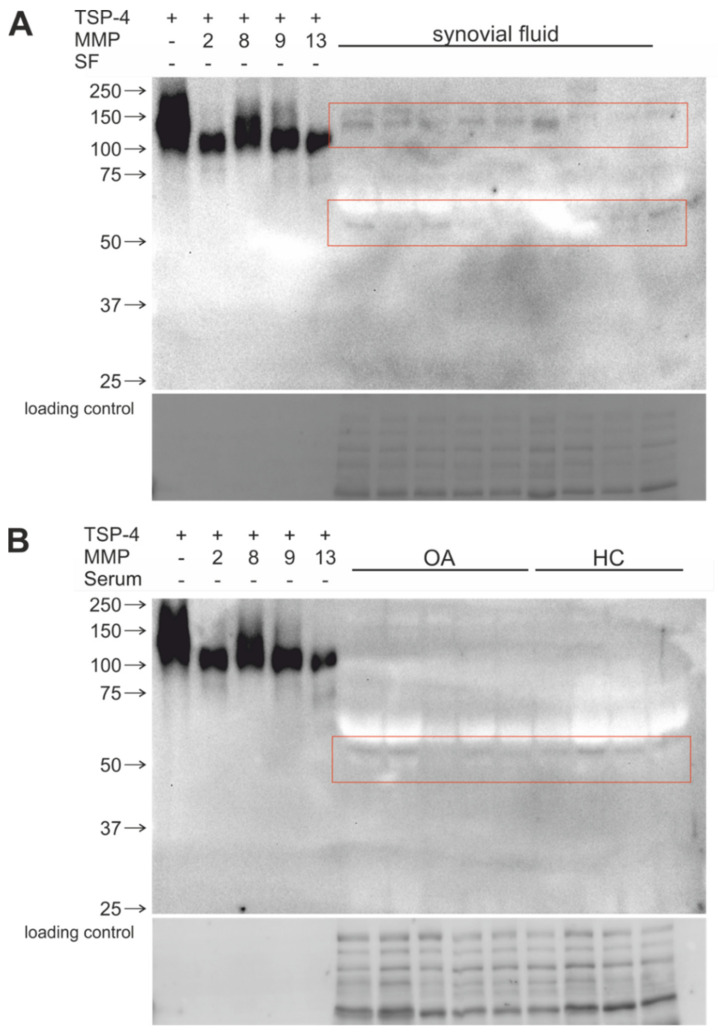
Comparison of in vitro MMP-generated TSP-4 fragments with cleavage products detected in human synovial fluid (**A**) or serum samples (**B**). In (**B**) serum samples from healthy individuals (HC) were included. Fragments were visualized with a polyclonal TSP-4 antibody. Red boxes indicate relevant protein fragments. (+) indicates with, (−) without substrate; protein sizes are shown in kDa. The lower panels in (**A**,**B**) display the loading control.

**Figure 9 ijms-23-06245-f009:**
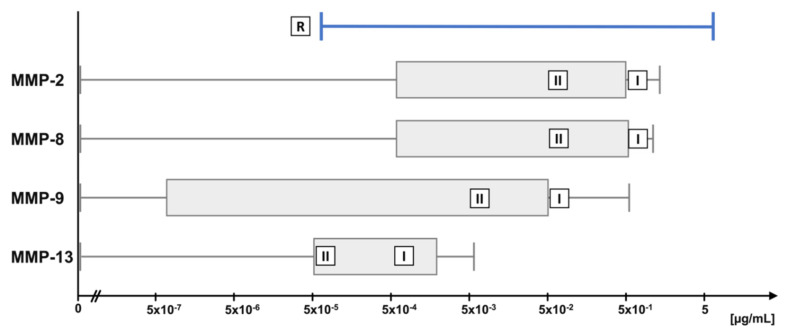
MMP-2, -8, -9, and -13 concentration ranges previously described in the literature (for references see text) depicted as median with inter-quartile ranges. MMP concentration ranges (R) used in the cleavage assays. Physiological lower (II) and ten times higher (I) concentrations were used for long-term digestion.

**Table 1 ijms-23-06245-t001:** MMP-concentrations used for long-term digestion.

MMP	Low Concentration (µg/mL)	High Concentration (µg/mL)
MMP-2	5 × 10^−2^	5 × 10^−1^
MMP-8	5 × 10^−2^	5 × 10^−1^
MMP-9	5 × 10^−3^	5 × 10^−2^
MMP-13	5 × 10^−5^	5 × 10^−4^

## Data Availability

Not applicable.

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
