# Peer review of "Generation of Matrix Degradation Products Using an In Vitro MMP Cleavage Assay"

_ijms, 2022, doi:10.3390/ijms23116245_

Round 1

Reviewer 1 Report

1) Line 17, "a biological activity" should be changed to "biological activities".

2) Line 18, "Further" should be changed to "Furthermore".

3) Line 57, "what cartilage matrix component" should be changed to "which cartilage matrix components".

4) Line 131, elaborate what "to a different extent" means.

5) Line 158, "it seems as if" should be changed to "it seems that".

6) There should be a discussion about possible reasons why MMPs cleave COMP more efficiently than TSP4.

7) Figures of the MMP/COMP/TSP4 structures should be included. These structures should be obtained from the PDB or AlphaFold. For example, the structure of TSP4 can be found here: https://alphafold.ebi.ac.uk/entry/P49744.

8) The authors should also perform protein-protein docking and include figures of COMP and TSP4 docked to the MMP. From the docking results, perhaps COMP interacts better with MMP, which might be why MMPs cleave COMP more efficiently. A discussion of protein-protein interactions should be included.

Reviewer 2 Report

Manuscript number: IJMS-1638864

Title: Generation of matrix degradation products using an in vitro 2 MMP cleavage assay.

This article by Dr. Niklas Wagner et al. reported interesting results about the cleavage products of cartilage oligomeric matrix protein (COMP) and thrombospondin-4 (TSP-4) by MMP-2, MMP-8, MMP-9, and MMP-13. It is an inspiring research paper to understand the roles of MMPs in the pathogenesis of osteoarthritis (OA). However, I would like to suggest that some concerns may help authors improve the manuscript.

Major

  1. The author purified proteins of interest such as COMP, TSP-4, and four pro-MMPs. In particular, the MMP activities are essential to elucidate the roles of MMPs, like the cleavage efficiency. Therefore, it would be great to validate and determine the purified MMPs activities using well-known fluorescent-based MMPs enzyme assay methods.

  1. The authors observed that fragments were generated from COMP or TSP-4 by MMP activities. Two questions are raised, one of which is incubation times. It may be duplicated question as above. Most MMP enzyme activity is a faster than few hundred pmoles/min/ug. Therefore, 1 ug of substrates, TSP-4 and COMP, will be digested very quickly within an hour. 24 hours incubation to investigate cleaved fragments is too harsh a condition. I would like to suggest repeating in vitro proteolysis assays shorter than 24 hours. The other concern was raised from patient samples. Authors claimed that “a number of different MMPs in a wide range of concentrations were detected in the synovial fluid and serum of OA patients.” Therefore, authors must determine four MMPs activities (2, 8, 9, and 13) in OA samples. In addition, investigating other proteases activities would be beneficial to understand the turnover of cartilage tissues (completely degradation of 70 and 60 kDa from COMP).

  1. Experiment design.

Authors isolated pro-MMPs for testing. The Pro-form of MMPs would be great a negative control for in vitro proteolysis assays. pan-MMP inhibitors (Ilomastat or Marimastat) can be used as a negative control. Align sequences of COMP and TSP-4 to find common cleavage sites by tested MMPs. Eventually, Edman degradation will be required to characterize the cleaved fragments.

Minor

  1. Remove “?” in line 47
  2. Replace large “A” to small “a” in line 69
  3. Add references for “different post-translational modification or to a proteolytic cleav-97 age as reported earlier for thrombospondins” in line 97
  4. Correct “activ atability” in line 99
  5. Relocating section 2.2 into the method.
  6. Correct “essay” to assay in line 160
  7. Remove duplicated reference in line 225
  8. The references (51 and 52) did not support the authors’ statement “however, others reported similar fragments. I could not find a result about TSP4 fragments by MMPs in both references. The reference 53 was added to support “further COMP-digesting proteases active” in line 267. The main topic of reference 53 is about TSP-4, not COMP. Please check it carefully.
  1. Please add gene entry ID and detail for cloning in 4.1.
  2. It would be better to add the gels of purified proteins to check the purity.
  3. Loading control is required to compare the amount of cleavage product among serum samples.

Round 2

Reviewer 1 Report

The authors made an argument for not including the structure although many papers have included docking results which more often than not, are very helpful. In any case, this paper is probably publishable based on merit of the experimental results.

Author Response

We thank the reviewer for the positive feedback. In response to the remaining comment, we added a few sentences in the discussion on the potential of structure prediction with Alphafold (lines 487ff). 

Reviewer 2 Report

Manuscript number: ijms-1638864

Title: Generation of matrix degradation products using an in vitro MMP cleavage assay

Dr. Niklas Wagner et al. revised all raised comments very well. However, I would like to suggest that some concerns may help the authors improve the manuscript. 

1. MMPs concentration in synovial fluid.

The authors selected the concentration of MMPs through a literature search. However, it would be great to show the actual concentration of MMPs if you save the synovial fluid samples.

2. In the supplement figure, an amount of COMP was not reduced by active MMPs. In addition, the size of COMP was raised in the presence of inactive all MMPs. Please revise the results and explain the gain of weight by inactive MMPs.

3. Revise Table 1.

Author Response

  1. MMPs concentration in synovial fluid.

The authors selected the concentration of MMPs through a literature search. However, it would be great to show the actual concentration of MMPs if you save the synovial fluid samples.

Response: We agree that it could be of interest to know the actual MMP concentration in synovial fluid samples. However, as noted by reviewer, we selected MMP concentrations based on a literature search which could be more representative to allow a general conclusion. Unfortunately, we also used up some of our samples to run additional gels as requested by the reviewer. Further, we are uncertain how long the MMP activtiy in synovial fluids remains stable upon long term storage. There might also be some self-digestion and we therefore suggest to determine MMP activities more or less immediately after sample collection.

We added in the discussion (line 396ff):

In the present study, we selected MMP concentrations based on a literature research. However, it could be interesting to determine the actual MMP concentration in individual samples and to correlate these concentration with fragmentation patterns. However, this should be done immedaiately after sample collection to avoid activity loss or self-digestion    

  1. In the supplement figure, an amount of COMP was not reduced by active MMPs. In addition, the size of COMP was raised in the presence of inactive all MMPs. Please revise the results and explain the gain of weight by inactive MMPs.

Response: We thank the reviewer for pointing us to this inconsistency. We provide a new supplemental figure 1 with more time points and a much clearer and hopefully convincing result. 

  1. Revise Table 1.

Response: We apologize for the formatting mistake and revised table 1 accordingly.